# Bioengineering of Antibody Fragments: Challenges and Opportunities

**DOI:** 10.3390/bioengineering10020122

**Published:** 2023-01-17

**Authors:** Sama Pirkalkhoran, Wiktoria Roksana Grabowska, Hamid Heidari Kashkoli, Reihaneh Mirhassani, David Guiliano, Colin Dolphin, Hanieh Khalili

**Affiliations:** 1School of Biomedical Science, University of West London, London W5 5RF, UK; 2Nivad Pharmed Salamat Company Research Center, Tehran 1455714138, Iran; 3School of Life Science, College of Liberal Arts and Sciences, University of Westminster, London W1W 6UW, UK; 4School of Pharmacy, University College London, London WC1N 1AX, UK

**Keywords:** bioengineering, biotherapeutics, antibody fragments, baculovirus expression system, plants expression system

## Abstract

Antibody fragments are used in the clinic as important therapeutic proteins for treatment of indications where better tissue penetration and less immunogenic molecules are needed. Several expression platforms have been employed for the production of these recombinant proteins, from which *E. coli* and CHO cell-based systems have emerged as the most promising hosts for higher expression. Because antibody fragments such as Fabs and scFvs are smaller than traditional antibody structures and do not require specific patterns of glycosylation decoration for therapeutic efficacy, it is possible to express them in systems with reduced post-translational modification capacity and high expression yield, for example, in plant and insect cell-based systems. In this review, we describe different bioengineering technologies along with their opportunities and difficulties to manufacture antibody fragments with consideration of stability, efficacy and safety for humans. There is still potential for a new production technology with a view of being simple, fast and cost-effective while maintaining the stability and efficacy of biotherapeutic fragments.

## 1. Introduction

Biotherapeutics or biologics are referred to the group of macromolecule drug products where the active substance is extracted or produced from a biological source [1]. Biotherapeutics include cytokines, growth factors, hormones, vaccines, proteins, and peptide-based products, as well as antibody-based medicines [2]. Monoclonal antibodies (mAbs) are considered the most rapidly growing biotherapeutics that have been used successfully for the treatment of chronic diseases such as cancer, inflammation, and ocular neovascularization [3,4,5]. IgG antibodies are mono-specific, bivalent molecules (Figure 1) with two Fabs (antigen-binding fragment) (Figure 1) and a fragment crystallizable (Fc) domain. While Fabs are responsible for selectively targeting cytokines or cell receptors, the Fc domain is required for both stability and Fc-mediated recycling, responsible for long circulation half-life in IgG. Stability also depends on the presence of intramolecular bonds within the light and the heavy chains and sugar groups located on the Fc fragment. Maintaining the stability of IgG antibody after being administrated to the patient, is the major concern for development of novel biotherapeutics because if it aggregates or degrades, it could cause unwanted immunogenicity in patients [6]. There are ongoing efforts to develop new classes of antibody-based medicines with a focus on increasing functionality and stability.

Antibody fragments such as Fab and scFv (single-chain variable fragment, Figure 1) are emerging biotherapeutic-based medicines for indications where smaller-sized molecules are required for better tissue penetration. Using smaller-sized fragments can improve potency by increasing the effective dose with a higher density of the target binding in a given volume [7]. Another potential advantage of antibody fragments is that their manufacture is relatively more straightforward in comparison to mAbs and less costly due to the lack of specific glycosylation requirements. These properties would permit the use of prokaryotic expression systems. Antigen binding fragments (Fabs) are the first class of antibody fragments with four FDA-approvals for different clinical applications, such as ocular neovascularisation and rheumatoid arthritis (Table 1). Single-chain fragments (scFv) are a new set of recombinant molecules in which the variable regions of light (V_L_) and heavy chains (V_H_) are produced as a single polypeptide joined by a flexible linker sequence. To enhance stability and binding affinity, amino acid sequences in V_H_ and V_L_ are modified. To date, only one scFv, brolucizumab, used for the treatment of age macular degeneration (AMD), has received FDA approval (2021). Because of its smaller size, brolucizumab can be administered at higher doses, resulting in a subsequent decrease in the frequency of intravitreal injection [8,9], compared to the currently approved drugs for treatment of age macular degeneration disease such as ranibizumab (a Fab molecule) and bevacizumab (a mAb). However, post-marketing concerns over safety and stability have been reported to the American Society of Retinal Specialists (ASRS), and case studies have subsequently been published [10]. Hence, there is still a need to manufacture the scFv to overcome the challenges associated with stability and safety.

From the pharmacokinetic perspective, the rapid clearance from the blood circulation due to their relatively small size and lack of an Fc domain, complicates the therapeutic use of antibody fragments. In addition, rapid degradation and instability are also major concerns and challenges for antibody fragments development. Different strategies have been developed to extend the half-life of antibody fragments and enhance their stability, such as conjugation to proteins (e.g., albumin [26]) or to polymer (e.g., PEGylation [26]). For example, certolizumab pegol (Cimzia, anti-TNFa PEG-Fab) is a PEGylated-Fab approved by the FDA for treatment of rheumatoid arthritis.

A variety of bioengineering techniques have been developed during antibody fragments manufacture with the aim of improving stability, circulation half-life and binding affinity. BiTEs (bispecific T-cell engager, Figure 1) are molecules generated by combining two scFvs via a peptide linker [27]. The peptide linker in the BiTEs is freely rotatable and flexible, which contributes to BiTE stability [28]. The two scFvs in BiTEs are derived from a different monospecific mAbs, which enable BiTEs to bind to two different targets resulting in enhanced functional and binding activity as compared to a single scFv. Blinatumomab (FDA approved in 2016), tebentafusp (FDA approved in January 2022) and solitomab (in clinical trials) are BiTEs biotherapeutics for the treatment of acute lymphoblastic leukemia, eye cancer and colon/lung cancer diseases, respectively.

There exist both challenges and potential opportunities in the production of stable and effective antibody fragments. This review will discuss both current bioengineering platforms used to manufacture antibody fragments and an overview of possible future avenues for generating fragments with enhanced efficacies.

## 2. Technologies to Bioengineer the Antibody Fragments (Fabs and scFvs)

While antibody fragments can be produced using enzymatic digestion of monoclonal antibodies, they can also be manufactured using recombinant heterologous protein expression systems. Host platforms such as bacteria, yeast, fungi, mammalian cells or even whole animals and plants have been successfully employed. While some of the biotherapeutics were originally extracted from human tissue and in relatively small amounts, the vast majority of therapeutic biologics on the market today are recombinant proteins, generated using reliable and consistent cell-based production platforms. Using recombinant DNA technologies, large amount of highly purified biologics can be produced, but challenges with the purity and quality still exist. In the following section, different bioengineering technologies are discussed, focusing on the advantages and disadvantages for production of antibody fragments.

### 2.1. Hybridoma Technology

Using hybridoma technology invented in 1975 [5], the first monoclonal antibody, muromonab-CD3 (OKT3), was designed and manufactured as a potential therapy to reduce kidney transplant rejection problems [5]. Several other monoclonal antibodies have been developed for research, diagnostic and as medicine for treatment using hybridoma technologies [29,30]. In hybridoma technology, rabbits or mice are immunized by administering antigen over the course of several weeks to strengthen the plasma and memory B cells of the immune system [29]. The activated B-cells (lymphocytes) are then isolated from the spleen using centrifugation, based on density gradients [31]. These cells are then fused with myeloma cells using an electric field [32,33] to generate hybridoma cells, which results in continuous production of an antibody specific for a single epitope. Hybridization only occurs under ideal conditions, and in the best of situations, only 1–2 percent of hybridoma cells (one hybrid cell per hundred) are eventually stable in culture to produce antibodies [31,34,35]. Despite the productivity and specificity of hybridoma-produced antibodies, the approach has some drawbacks such as the long process of developing and cloning hybridoma cells, the murine origin of the antibodies, and limited control of epitope differentiation. For example, approximately 50% of patients treated with OKT3 elicit a human anti-mouse antibody (HAMA) reaction [32,33] even after just a single dose. Such HAMA effects interfere with the binding of OKT3 to T-cells and results in reduced therapeutic efficacy [33]. In addition, the effectiveness of murine-based antibodies is limited because they tended to display short half-lives (less than 20 h), elicit high immunogenicity and exhibit ‘suboptimal’ effector functions. Hybridoma technology has also been used to generate scFvs, but the approach resulted in the heterogonous products because more than one heavy and light chain was produced from one cell line [31]. Other technologies allowing development and production of fully human or humanized homogenous scFvs/Fab biotherapeutics are now widely employed to address these issues and enhance efficacy.

### 2.2. Phage Display

Phage display was used to produce the first fully human antibody, adalimumab. It uses engineered bacteriophage (a virus-infecting bacteria), repeated rounds of antigen-guided selection and phage propagation [36] that can successfully solve the problem of non-humanized monoclonal antibodies. These phage display libraries include human variable heavy and light chains, allowing amplification of all transcribed rearranged variable regions within a given immunoglobulin repertoire during library construction. Combinatorial libraries of antibody V_H_ and V_L_ genes are produced by expressing them on the surface of phages. These libraries can then be used to generate antibodies [37]. The cDNA libraries are then produced using different cell’s mRNAs which inserted into phage [38]. After phage panning, phage containing the desired gene can be cloned into competent *E. coli* strains to obtain more vectors containing specific antibodies or into *E. coli* expression strains to get pure recombinant proteins. To determine whether expressed antibodies have any binding toward their specific targets, surface plasmon resonance (SPR) methods are suggested [39]. This technique is not only capable of measuring the binding affinity between analyte and ligand, it can also study the dynamic interaction (binding association and dissociation) of the produced antibodies toward its antigen in real time [40]. The SPR technique has advantages over other methods (e.g., ELISA and co-immunoprecipitation or Co-IP), as it is able to determine the binding affinity without the need of labeling a ligand or analyte and is capable to measure association rate contacts in real time [41]. Different types of libraries for different groups of biotherapeutics (e.g., scFV or Fabs) could be designed by phage display technology. A detailed discussion of different phage display libraries and production methods is beyond the scope of this review, and can be found in these reviews [41,42,43,44,45,46]. This technology, however, suffers from the method’s complexity, the high cost and the fact that it is very time-consuming to generate a library. In addition to this, misfolding of the heavy and light chains can occur, which results in the generation of non-active antibody fragments [47].

### 2.3. Transgenic Animals

Because the phage display method suffers from experimental complexity, other methods have been used to produce fully human antibodies such as transgenic animal. Transgenic mice with integrated human immunoglobulin loci [48,49,50,51] was developed to aid in the rapid generation of human biotherapeutics. The human immunoglobulin loci include the repeated and highly homogenous sets of genes was rearranged in mouse B-cells to produce an individual’s highly variable repertoire of expressed antibody, which was then designed to bind to a specific antigen. In the transgenic approach, natural diversification and selection is exploited, as integrated loci are under the control of the animal’s immune system, where they can undergo normal processes of DNA rearrangement, hypermutation and B cell selection [52].

The production of transgenic animals necessitates the utilization of a number of techniques, each of which entails injecting the recombined genome into the animals. Microinjecting DNA into embryo pronuclei using lentiviral vectors or transposons is one method of transferring the recombinant genome. Another method of transferring the recombinant genome includes incubating sperm with DNA followed by in vitro fertilization to obtain Intracytoplasmic Sperm Injection (ICSI) [53,54,55]. Microinjection was used to transfer ICSI into pronuclei; however, these vectors have a limited capacity to harbor foreign DNA because the number of integration sites in the same animal is limited and difficult to control. This strategy is not effective when used in ruminants, and has applied for mice, rats, rabbits, pigs and fish. Several efforts were made [56] to increase the frequency of genetic integration by inserting foreign genes into vectors and using vectors known as transposons and lentiviral vectors to overcome the issue stated above, explained in details in these reviews [57,58,59,60].

Pigs and mice are used as the experimental animals of choice when ICSI methods applied to produce a substantial number of transgenic animals [61]. The possibility that the process of transgenesis could, at least in theory, be streamlined is something that, in some other contexts, could be considered a drawback. One that is more effective is a method that, first, requires breaching the sperm membrane, second, involves incubating the sperm while they are in the presence of DNA, and third, involves fertilizing the oocytes with the use of ICSI [56,62]

For close to twenty years at this point, mice have been used for the purpose of extracting genes from their cells in order to facilitate the transfer of genes from other species. In spite of the fact that it is more straightforward, there is a possibility that this approach will render some genes inactive (gene knock out). In this scenario, pluripotent cells, which are cells that can participate in the production of chimeric transgenic animals, are utilized. These cells have the potential to become any type of cell with potential to develop into a variety of different organisms [60].

Gene targeting is an additional method that may be used to integrate foreign genes into genomic regions using gene targeting. This methodology employs homologous recombination as the way by which it achieves both the insertion of genes and the targeting of certain genes. As a result, the method is capable of attaining both of these objectives. Gene knockout is one of the other options that can be utilized when carrying out this operation [62]. The main challenge associated with transgenic animal is cost and time required to employ transgenic animal to create antibody. Transgenic cattle were also proposed for high production of bispecific scFv for treatment of human melanoma [52], but challenges to maintain stability and safety are still applied.

### 2.4. Single B-Cell Technology

Single B-cell technology is using normal or immunized human donors to engineer monoclonal antibody in situ. This technology is based on the fusion and immortalization of human B-cell with the Epstein–Barr virus [52,63], which turn healthy B-cell into lymphoma B-cell [64]. The vital advantage of this method is that it allows the isolation and formation of native human antibodies with the natural pairing of V_H_ and V_L_ [64], which is not possible in phage display and transgenic animal technologies. There are different methods for screening single cells that express desired antibodies, such as fluorescent-activated cell sorting (FACS), micro-engraving and fluorescent-activated droplet sorting (FADS) [65]. Here, we describe some of these screening methods in details.

Fluorescent-activated cell sorting is a specialized type of flow cytometry that was developed in the 1960s [66]. It involves suspending and vibrating cells that have been stained with reagents (antibodies and antigens) that were previously tagged with fluorescent molecules. Labeled cells are further divided into different groups based on their specific fluorescent properties, which aid them to move toward a measuring station. Among all other screening methods, FACS has the benefit of being feasible to enrich a population of desired single B cells more quickly and thoroughly. The chosen population is divided further depending on charge (positive, negative or neutral) to distinguish between several phenotypes of a particular antibody [67].

Micro-engraving is another approach for screening of single cell which used micro-engraved microwells containing up to 62,500 polydimethylsiloxane (PDMS) wells with volumes ranging from 0.1 to 10 nL. Cells are seeded into a chip with micro-engraved wells, and are analyzed using a specific labeled antigen or antibody. Following the selection of cells that produce an antigen-specific antibody, the V_H_ and V_L_ of the desired antibody gene sequence will be examined using either a micropipette for clonal expansion or recombinant retrieval or by lysing cells and performing RT-PCR [68].

Another method for isolating a single B cell in nano amounts is microarray screening (also known as fluorescent activated droplet sorting FADS). This approach, similar to FACS, employs microfluidic chips to detect the presence of a specific product in droplets [69]. This technology could overcome some of the limitations associated with FACS, such as the ability to analyze secreted proteins that are present in droplets, making it a functional method for isolating single B cell for specific enzymes or cytokines [69,70].

Anti-Candida monoclonal antibodies, which target yeast infections (the genus Candida), are an example of one of the monoclonal antibodies created using B-cell technology. These monoclonal antibodies are designed to have improved phagocytic properties for the treatment of Candida infection diseases. Additionally, the single B-cell technique has proven extremely effective in creating anti-viral monoclonal antibodies, including those that are specific for the rotavirus and the cytomegalovirus pp65 antigen in humans [71]. However, this approach has certain limitations, such as stability concerns for fused B-cells and not being applicable for all treatments [72].

## 3. Heterologous Protein Expression Platforms for Antibody and Antibody Fragment Production

In recombinant DNA technology, different hosts or expression systems are used to produce different biotherapeutics ranging from full antibody to antibody fragments. The main advantage of this technology is the ability to manipulate the gene of interest to produce the antibody. In contrast to other methods, using recombinant DNA technology could result in high yield expression of specific target antibodies in hosts because it is possible to monitor and optimize all the process in the culture media [73]. In the following section, we will focus on different expression systems to engineer Fabs and scFvs using DNA recombinant technologies.

### 3.1. Bacterial Expression (E. coli)

*E. coli* was the first host used for producing antibody fragments because of its potential to provide a more commercially viable high-expression and low-cost platform [74,75]. The two most common methods for protein secretion in *E. coli* are via utilization of the Secretion (Sec) pathway or the twin arginine translocation route, which are explained in brief here. The Secretion pathway is the major transport route for proteins that are exported from the cytoplasm in bacteria [76]. Generally, secretion substrates are created in the form of larger molecular weight precursor proteins that are accompanied by an amino-terminal signal peptide [77]. This signal peptide directs protein to the membrane-bound secretion translocase. During the synthesis of secretory proteins, the signal recognition particle (SPR) recognizes the precursor proteins via their extremely hydrophobic signal peptide [78]. After protein synthesis, precursor proteins with fewer hydrophobic signal peptides interact with post translationally interacting proteins (PiPs) [79]. These proteins protect secretory proteins from aggregation and keep them in an unfolded state that is suitable for secretion [80,81]. The signal peptide is broken down by signal peptidase either during membrane translocation or shortly after [82]. This allows the mature protein to be released on the trans side of the membrane.

The second secretion method is based on the presence of a twin-arginine pair in the signal peptides. Most bacterial species make use of a system that has been named the twin-arginine translocation route, also known as Tat for short. This delivery rout results in the protein being secreted in folded, and in some cases even oligomeric [83]. In Gram-negative bacteria with a high GC-content, the twin-arginine translocase is composed of the components TatA, TatB and TatC; however, in Gram-positive bacteria with a low GC-content, a minimal translocase that only consists of TatA and TatC is active [84,85,86]. Fully folded precursor proteins bind to a substrate receptor in the cytoplasmic membrane that is formed by TatB (or, alternatively, a bifunctional TatA protein) and TatC [87,88,89]. After synthesis and folding in the cytoplasm, which frequently involves the insertion of a tightly or even covalently bound cofactor, precursor proteins bind to the receptor. The cytoplasmic folding reaction is the name given to this phenomenon. Again, the signal peptide is cleaved by signal peptidase, and after, the mature protein is released on the trans-side of the membrane [90].

The *E. coli* expression method in general suffers from drawbacks including limited scope for secretion into the culture medium and the lack of protein glycosylation. In addition, there is the potential for intracellular aggregation of highly expressed recombinant proteins, resulting in inclusion body formation and the production of misfolded protein [91]. Endotoxins produced during bacterial expression, difficult to remove, is another limiting factor associated with the *E. coli* expression system [91].

In regards to monoclonal antibodies, *E. coli* is not a suitable expression system because of its inability to glycosylate antibodies in their Fc fragment. The endoplasmic reticulum and Golgi apparatus, present in most eukaryotic but not prokaryote cells, are responsible for the glycosylation [91,92]. Unlike larger antibody subunits, single chain variable fragments and antibody fragments do not need to be glycosylated. Thus, *E. coli* could be used as suitable expression platform. A high yield of Fab (over 4.0 g/L) and single-chain variable fragments (up to 3.5 g/L) has been achieved by using the recently developed ESETEC secretion technology (Wacker Biotech) to secrete recombinant products into the broth culture during fermentation [93,94]. Limitation with protein folding and production of biologically active biotherapeutics has led to develop mammalian cell lines as a preferred method [92].

### 3.2. Mammalian Cell Lines

Over 60% of all biotherapeutic products, including monoclonal antibodies, are produced using mammalian expression systems. This is mainly due to the ability of mammalian cells to synthesize and fold proteins in a similar manner to humans. The ability of these cells to also add post-translational modifications (PTMs) such as glycans, which aid with proper folding, stability, and activity, is a key feature driving the success of these platforms [95] and currently distinguishes them from all other recombinant protein production systems. Because glycosylation patterns can influence the stability or function of the product, and because non-natural glycoforms may be immunogenic, glycoform profiles hold considerable weight in the realm of protein pharmaceuticals. Proteins that are expressed in conventional rodent cell lines often harbor several terminal glycan epitopes, such as N-glycolylneuraminic acid (NGNA) or galactose-a-1,3 galactose (a-Gal), not found in human glycoproteins. These have the capacity to elicit an immune response in patients, which can compromise therapeutic efficacy via enhanced clearance or, in more rare instances, cause serious side effects [96]. An example of this is the commercial antibody Erbitux (cetuximab), which was produced in a cell line derived from a murine myeloma that added a-Gal epitopes and caused development of an IgE-mediated anaphylaxis reaction in some patients [97].

It was previously believed that some rodent cell lines such as CHO cells did not have the necessary biosynthetic machinery to produce an a-Gal epitope [98]; however, glycan profiling of Orencia (abatacept) demonstrated that some clones of these cells are capable of producing a-Gal containing products [96]. As a consequence of this issue and the high genetic variability in many rodent cell lines, the glycosylation profiles of the specific clones that are utilized in the expression of recombinant proteins needs to be analyzed to guarantee that they are consistent and free of a-Gal and other non-human glycan product. Because of the high genetic diversity of CHO and other rodent cells and the availability of genes that are functionally hemizygous [99,100], it is possible to engineer lines that either lack specific glycosylation activities or produce enzymes resulting in humanized glycan profiles. Specific CHO cell lines have been engineered to either recapitulate human glycosylation patterns or produce glycoforms with enhanced therapeutic activities [101]. As these mutant or engineered cell lines require certain nutrients or selection markers to be added to their growth media, they offer a more stable progenitor for the isolation of producer lines [102].

In some instances, the productivity of mammalian cells cultivated in bioreactors has reached very high levels, i.e., 10–15 g/L in monoclonal antibodies and Fc-fusion protein production [103]. However, generally, they give much more lower yields and mammalian expression systems, present some common challenges, such as the need of expensive production platforms and stringent and consistent cell growth conditions to maintain cell viability, and prevent the release of byproducts into the media during manufacturing. For example, variations in culture conditions can result in drastic variations in the PTMs and functional activity of final products [101]. Hence, demonstrating a detailed understanding and stringent control over process parameters is critical in getting the required quality attributes from these cell lines. This can be challenging, especially during the implementation of the process into a new lab and during the scaling up process. Nevertheless, different cell lines, such as Chinese Hamster Ovary (CHO), Human Embryonic Kidney (HEK) and Cellosaurus NS0 (Group: Hybridoma fusion partner cell line), have been successfully used as high expression and yield host cells in the industry. Idarucizumab, approved in 2015, is an example of a Fab produced with a CHO cell expression system. Blinatumomab is also expressed in CHO cells. Developing a stable mammalian cell line is a crucial step for ensuring its usage in the large-scale production. This can be achieved through the insertion of the gene of interest (GOI) into the random segments of the genome and then selection of high-producing, stable cells from the polyclonal pool of cell lines. Different selective agents have been applied to generate stable cell lines with high levels of expression [103]. Dihydrofolate reductase (DHFR), a vital gene for cell growth and proliferation, for instance, has been employed as a marker for CHO cell selection and amplification [100]. Additionally, methotrexate treatment, can be used for the amplification of expression cassettes, is another approach for establishing high-expression stable cells [103]. Development of stable cell line with high production yield and desired PTMs is, however, very time consuming and could take up to several years of optimization. However, for full antibody production [104] and high throughput screening [105], mammalian cell expression systems are still preferred methods because of its high production yields, product stability and possible glycosylation [106]. In terms of antibody fragments, other expression systems, such as plant and insect cell-based systems, as described below, could have potential advantages over mammalian cell system because of their simplicity and ease of use for high expression of pure, active and stable antibody fragments.

### 3.3. Plant-Based Expression Systems

Transgenic plant expression system has been used for several years to produce recombinant proteins, such as enzymes and antibodies. Recently, this expression system has been emerged as an alternative for production of antibody fragments, for example single-chain Fv antibodies (scFv800E6) against an ErbB-2, which were successfully expressed in nicotiana tabacum [107]. Selection of the specific plants (e.g., Solanum tubersum, Nicotiana tabacum and Nicotiana benthamiana) to express desired proteins, play an important role in a protein’s efficacy and stability [108]. Expressed protein can be secreted into extracellular space or retained in the Endoplasmic Reticulum (ER) [108]. The remaining scFvs in ER space have been suggested to aid with the proper folding (using different molecular chaperons present in ER) and stability of the proteins, as aggregation, self-degradation and digestion in extracellular space could be prevented [109]. The functional activity of the scFv produced in transgenic plants was similar to what was generated in either bacteria or yeast, but challenges for protein extraction and purification from plants had an impact on the yield of the final product [110].

In general, the transgenic plant expression system has some advantages, including great flexibility in production yield and no need for having expensive cell culture facilities, as required for the mammalian system [109]. There are various downsides to employ plants as expression platforms, as they are limited in the types of glycosylation they can perform and may in some instances have low expression yields. Selection of plants for high expression yield can also be challenging; additionally, some allergic reactions to antibodies produced by plants have been observed [111].

### 3.4. Insect Cell Expression System

Production of recombinant proteins in insect cells has a long history dating back to the mid-1980s. The baculovirus expression vector system (BEVS) is the most popular example of this platform. Baculoviruses mediate natural viral infections of insect cells and are generally not considered hazardous to humans. Several gene cassettes can be introduced into the double-strand DNA genome either via in vivo homologous recombination between a shuttle vector and linearized genome [112] or specific transposon-mediated insertion of the desired sequence from a vector into the insertion site in a single copy of the BV genome (bacmid) maintained within an *E. coli* host (bacmid) [113].

Protein expression level depends on the nature of the expressed protein. Frequently, expression of proteins in the cytoplasm is higher than for proteins engineered to be secreted into the media. Co-transfection with baculoviruses expressing chaperone proteins, to prevent protein aggregation, is employed to increase the synthesis of functional proteins [111,114].

Unlike bacterial expression systems, insect cells are capable of most posttranslational changes observed in mammalian cells. N-glycans contribute significantly the utility of glycoproteins in a variety of ways, including immunogenicity and biological activity [115,116].

N-linked glycosylation in insect cells results in formation of glycoproteins with simple oligo-mannose sugar chains [117]. Mammalian cells, on the other hand, form glycoproteins with complex sugar groups and terminal sialic acids [117,118,119]. Hence, the value of these recombinant glycoproteins is limited due to inconsistencies between expected and actual glycosylation patterns, particularly the absence of terminal sialic acids. When compared to the original mammalian protein, these changes influence the protein’s immunogenicity as well as its biological features [120]. To overcome this limitation of the baculovirus expression technique, researchers have attempted to engineer insect cell lines that express the additional enzymes required for the formation of mammalian glycosylation patterns [117,121,122,123]

Insect cells are cultured at 25–28 °C without of need CO_2_. The ease of the culture conditions needed to grow insect cells is one of the major advantages of this expression system compared to mammalian cells. After culturing, cells are infected with the recombinant baculovirus carrying the gene of interest [113]. Since the baculovirus–host interaction is of a lytic nature and ends with insect cell death, recombinant proteins could be produced in batch cultures and continuous production of protein is not impossible [113]. An alternative to overcome this problem is the use of transformed insect cells for the continuous production of recombinant protein [124,125,126]. A key selling point of this system is the ability to use it as a universal “plug and play” process, in which one requires to only change the recombinant baculovirus to produce a broad range of proteins [127]. This drastically reduces cell line and process development times, allowing production of various proteins using the same platform. Hence, using insect cell expression system could potentially save time in the production of biotherapeutics and provide some post-translational modifications.

## 4. Challenges and Opportunities of Different Expression Systems to Produce Antibody Fragments/Future Perspective

Monoclonal antibodies and their fragments have emerged as a major class of therapeutic agents with broad clinical applications. Technologies to manufacture antibodies have had a continuous evolution over the past 50 years but there is still a need for new technologies to address challenges in selectivity, stability, higher efficacy, reduced immunogenicity and side effects in humans. Costs of production is another key element toward a new technology because it would impact therapy expenses, which in turn has a huge pharmaco-economic burden for both patients and pharmaceutical companies.

While antibody engineering began with hybridoma technology, this method has suffered from low-yielding efficiency, cost and animal immunization, leading to the antibody sequence originated from an animal with potential possibility to trigger human immune response. Beyond hybridoma, other technologies emerged to produce “human-like” biotherapeutics, such as phage display, but their utility has been limited due to the complexity of phage library preparation. Recombinant DNA methods have been a very successful technology for production of antibody fragments, but the selection of an efficient expression systems can be challenging.

Figure 2 summarizes the challenges and opportunities that are associated with different expression platforms in regards to antibody fragment production. Expression systems are categorized into four main systems: bacterial, mammalian, plants and viral (insect cells).

Bacteria expression systems are low cost capable of producing a high yield of the majority of simple proteins and peptides that are not glycosylated. Because the recombinant vector may be injected directly into the bacterial host without virus contamination by humans and bacteria, there is no requirement for any additional stage of viral eradication. Bacteria on the other hand are unable to perform posttranslational protein modifications, such as glycosylation, g-carboxylation, phosphorylation and sulfitation [127]. Proteins that are inactive but have the correct molecular structure can also be produced by bacteria, which can lead to the formation of inclusion bodies. In addition, bacteria are capable of removing signal peptides, but they are unable to degrade preproteins, such as native coagulation growth factors [127]. Endotoxins, which are produced by bacteria during the expression, are notoriously difficult to eradicate and make the purification process difficult. While it is possible to use tagged purification method (e.g., his-tag) to isolate expressed antibody fragment, the presence of endotoxins and inclusion bodies challenges the purity of final product.

Mammalian expression systems are used for the most antibodies to manufacture glycosylated and rather complex protein structures. When compared to other platforms, the mammalian system is the only system that is capable of producing human native protein while performing the PTMs. This system, however, suffers from costly and complex procedure. Other significant drawback is the time-consuming approach, which results in a low output yield. In regard to purification, the removal of viruses is a necessary step in the purification process, which adds a level of complexity to the process.

Among other expression systems, plant expression systems offer a number of benefits, including low production costs and high protein yields. Despite the fact that recombinant plants are capable of accurately folding complex proteins and conducting more PTMs, the glycosylation profile of these plants is distinct from that of the native human equivalents, which has an impact on the immune responses of patients. The ability of transgenic plants to synthesize all proteins is hindered as a result of this drawback. It is possible for foreign proteins to be stored in the plant’s leaves, seeds or both, depending on the promoter that was used. Even though there are a lot of leaves, it could be difficult to isolate the protein of interest from them since they contain substances such as polyphenols and proteases, which patients do not tolerate very well [127].

Because of its low cost and the fact that it can perform most PTMs in a structure that is similar to that of humans, the baculovirus expression system is an excellent platform for the production of increasingly complex proteins. When compared to other systems, the flexibility and capability of plug-and-play production in baculovirus-insect cells are truly remarkable. Changing the target protein in one system will result in changes to all phases of production in the other systems; however, in the baculovirus-insect cell expression system, changing the target genome and inserting a new baculovirus backbone will result in new protein production.

While using expression systems such as insect cells and plants is still in a preliminary stage in both research and clinical development, we anticipate that they could revolutionize biotherapeutic fragments engineering because of their potential to be fast and more cost effective.

## Figures and Tables

**Figure 1 bioengineering-10-00122-f001:**
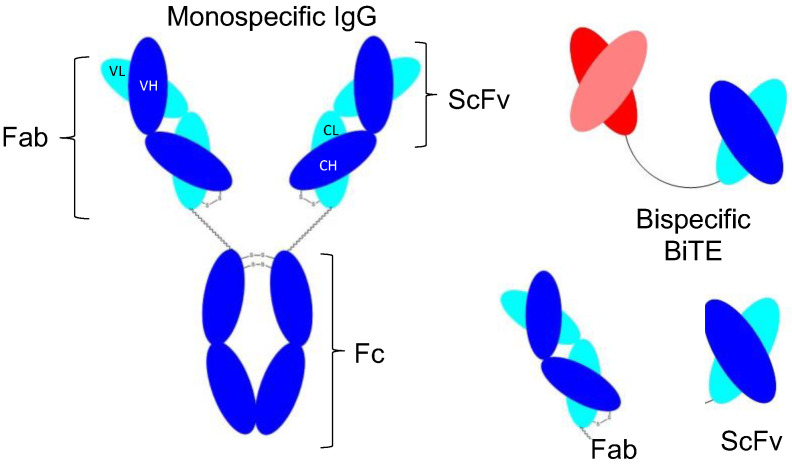
Structure of monospecific IgG, Fab, scFv and BiTE. V_H_: Variable Heavy Chain, V_L_: Variable Light Chain, C_H_: Constant Heavy Chain, C_L_: Constant Light Chain.

**Figure 2 bioengineering-10-00122-f002:**
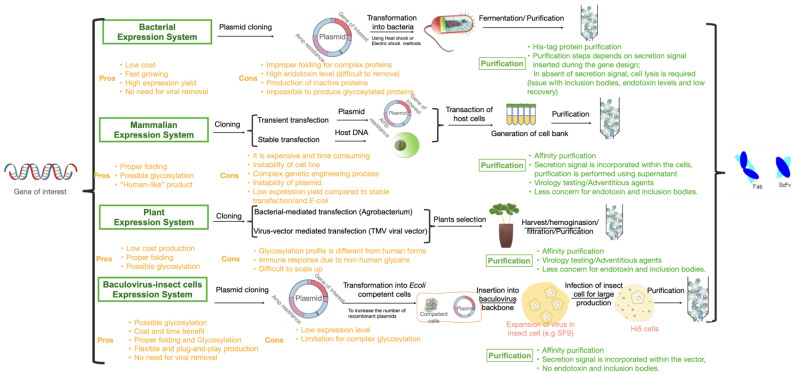
Challenges and opportunities for different expression systems to bioengineer Fabs and scFvs.

**Table 1 bioengineering-10-00122-t001:** List of Fabs and scFvs used in the clinic (or late stage of clinical development) with their specificity, targets and expression systems.

Molecule Type	International Non-Proprietary Name	Target	Format	Specificity	Sequence Source	Identification	Expression System	References
Single Chain Fragment (scFV)	Tebentafusp	gp100, CD3	TCR-scFv fusion protein	Bispecific	Humanized	Metastatic uveal melanoma	*E. coli* Bacteria	[11,12]
	Brolucizumab	VEGF-A	scFv	Monospecific	Humanized	Necvascular age-related macular degeneration	*E. coli* Bacteria	[13,14]
	Blinatumomab	CD19, CD3	BiTE scFv	Bispecific	Murine	Acute lymphoblastic leukemia	Chinese hamster ovary (CHO) cells	[15,16,17]
	Solitomab	CD3, EpCAM	BiTE scFv	Bispecific	Murine	Multiple solid tumors expressing EpCAM	Chinese hamster ovary (CHO) cells	[18,19]
Fab	Idarucizumab	Dabigatran Exilate	Fab	Monospecific	Humanized	Reversal of dabigatran-induced anticoagulation	Chinese hamster ovary (CHO) cells	[17,20,21]
	Certolizumab pegol	TNF	PEGylated Fab	Monospecific	Humanized	Crohn disease, Active Rheumatoid Arthritis, Psoriatic Arthritis	*E. coli* Bacteria	[22,23]
	Ranibizumab	VEGF	Fab	Monospecific	Humanized	Macular degeneration	*E. coli* Bacteria	[24]
	Abciximab	GPIIb/IIIa	Fab	Monospecific	Chimeric mouse/human	Prevention of blood clots in angioplasty	Murine myeloma cells (Sp2/0)	[17,25]

## Data Availability

Data available in a publicly accessible repository that does not issue DOIs.

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
