# Peer review of "Bioengineering of Antibody Fragments: Challenges and Opportunities"

_bioengineering, 2023, doi:10.3390/bioengineering10020122_

Round 1

Reviewer 1 Report

The manuscript under consideration is of great interest to the readers of the journal and provides a useful review of the literature.

The authors considered a wide range of technologies for antibody engineering, which certainly reflects the high scientific level of the work.

At the moment, the review rather falls into the category of mini-reviews, because each part is presented very concisely, literally 1-2 paragraphs using 4-6 references.

Before accepting  for publication, I would highly recommend significant expanding each section, enriching it with a comparative analysis.

It is absolutely necessary to generously discuss the advantages and disadvantages of all methods in a separate section, to determine the vector of development of each approach. Now this part of the review is presented very briefly, in the form of a listing. I would like to read the opinion of the authors about which areas will develop faster and which ones will stagnate.

Thus, the section on future prospects is also recommended to be significantly improved and expanded.

Author Response

Response: We are thankful for the Reviewer ‘s comment and we have now revised the manuscript extensively by expanding each section with more comparative analysis, including more examples /details for each expression systems, and adding a separate section on “challenges and opportunities for different expression systems”. We have merged section in “future perspective” with section in “challenges and opportunities” for better explanation. We have also added another 65 references to significantly improve the manuscript. Our main goal in the submitted manuscript is to compare the current bioengineering methods to produce safe, stable and active antibody fragments in a very concise and get to the point way.

Reviewer 2 Report

This MS is a short review on the R&D activities for developing antibody fragments, primarily scFv and Fab, to replace whole antibody molecules. Focusing on the bioengineering efficacy and the fragments' properties, they reviewed various expression systems for their unique advantages and disadvantages.

It is a well written and a concise MS; however, to further improve its quality, I suggest reinforcing "references" to the following points:

1. [Line #195-197] It is a well known fact, but a few representative references need to be listed for the interested readers.

2. [Line #266-267] The major concerns regarding E. coli systems are presented without proposing solutions for each concern. It is recommended to add some references for each so that the interested readers can further look into.'

Overall, I recommend "Minor Revision."

Author Response

Comment:

  1. [Line #195-197] It is a well known fact, but a few representative references need to be listed for the interested readers.

Response: We thank the Reviewer for the comments and we have now added references [81-88] for the content explained in Line #357-386

 in the revised manuscript

Comment:

  1. [Line #266-267] The major concerns regardingE. colisystems are presented without proposing solutions for each concern. It is recommended to add some references for each so that the interested readers can further look into.'

Response: We thank the Reviewer for pointing out this and we have now added few lines in page 6 and 7 in revised manuscript with some suggestions to address some issue associated with E-coli system and also added references [61-79] to improve the manuscript.

Round 2

Reviewer 1 Report

The authors have significantly improved the  manuscript. 

I recommend accepting in present form